# Nearest Neighbour Radial Basis Function Solvers for Deep Neural Networks

## Abstract

We present a radial basis function solver for convolutional neural networks that can be directly applied to both distance metric learning and classification problems. Our method treats all training features from a deep neural network as radial basis function centres and computes loss by summing the influence of a feature's nearby centres in the embedding space. Having a radial basis function centred on each training feature is made scalable by treating it as an approximate nearest neighbour search problem. End-to-end learning of the network and solver is carried out, mapping high dimensional features into clusters of the same class. This results in a well formed embedding space, where semantically related instances are likely to be located near one another, regardless of whether or not the network was trained on those classes. The same loss function is used for both the metric learning and classification problems. We show that our radial basis function solver outperforms state-of-the-art embedding approaches on the Stanford Cars196 and CUB-200-2011 datasets. Additionally, we show that when used as a classifier, our method outperforms a conventional softmax classifier on the CUB-200-2011, Stanford Cars196, Oxford 102 Flowers and Leafsnap fine-grained classification datasets.

## 1 Introduction

The solver of a neural network is vital to its performance, as it defines the objective and drives the learning. We define a solver as the layers of the network that are aware of the class labels of the data. In the domain of image classification, a softmax solver is conventionally used to transform activations into a distribution across class labels (Krizhevsky et al., 2012; Simonyan & Zisserman, 2014; Szegedy et al., 2015; He et al., 2016). While in the domain of distance metric learning, a Siamese (Chopra et al., 2005) or triplet (Hoffer & Ailon, 2015; Schroff et al., 2015; Kumar et al., 2017) solver, with contrastive or hinge loss, is commonly used to pull embeddings of the same class together and push embeddings of different classes apart. The two tasks of classification and metric learning are related but distinct. Conventional classification learning is generally used when the objective is to associate data with a pre-defined set of classes and there is sufficient data to train or fine-tune a network to do so. Distance metric learning, or embedding space building, aims to learn an embedding space where samples with similar semantic meaning are located near one another. Applications for learning such effective embeddings include transfer learning, retrieval, clustering and weakly supervised or self-supervised learning.

In this paper, we present a deep neural network solver that can be applied to both embedding space building and classification problems. The solver defines training features in the embedding space as radial basis function (RBF) centres, which are used to push or pull features in a local neighbourhood, depending on the labels of the associated training samples. The same loss function is used for both classification and metric learning problems. This means that a network trained for the classification task results in feature embeddings of the same class being located near one another and similarly, a network trained for metric learning results in feature embeddings that can be well classified by our RBF solver. Fast approximate nearest neighbour search is used to provide an efficient and scalable solution.

The best success on embedding building tasks has been achieved by deep metric learning methods (Hoffer & Ailon, 2015; Schroff et al., 2015; Song et al., 2016a; Sohn, 2016; Kumar et al., 2017), which make use of deep neural networks. Such approaches may indiscriminately pull samples of

the same class together, regardless of whether the two samples were already within well defined local clusters of like samples. These methods aim to form a single cluster per class. In contrast, our approach pushes a feature around the embedding space based only on the local neighbourhood of that feature. This means that the current structure of the space is considered, allowing multiple clusters to form for a single class, if that is appropriate. Our radial basis function solver is able to learn embeddings that result in samples of similar semantic meaning being located near one another. Our experiments show that the RBF solver is able to do this better than existing deep metric learning methods.

Softmax solvers have been a mainstay of the standard classification problem (Krizhevsky et al., 2012; Simonyan & Zisserman, 2014; Szegedy et al., 2015; He et al., 2016). Such an approach is inefficient as classes must be axis-aligned and the number of classes is baked into the network. Our RBF approach is free to position clusters such that the intrinsic structure of the data can be better represented. This may involve multiple clusters forming for a single class. The nearest neighbour RBF solver outperforms conventional softmax solvers in our experiments and provides additional adaptability and flexibility, as new classes can be added to the problem with no updates to the network weights required to obtain reasonable results. This performance improvement is obtained despite smaller model capacity. The RBF solver by its very nature is a classifier, but learns the classification problem in the exact same way it learns the embedding space building problem.

The main advantages of our novel radial basis function solver for neural networks can be summarised as follows:

- Our solver can be directly applied to two previously separate problems; classification and embedding space learning.
- End-to-end learning can be made scalable by leveraging fast approximate nearest neighbour search (as seen in Section 3.2).
- Our approach outperforms current state-of-the-art deep metric learning algorithms on the Stanford Cars196 and CUB-200-2011 datasets (as seen in Section 4.1).
- Finally, our radial basis function classifier outperforms a conventional softmax classifier on the fine-grained classification datasets CUB-200-2011, Stanford Cars196, Oxford 102 Flowers and Leafsnap (as seen in Section 4.2).

## 2 RELATED WORK

**Radial Basis Functions in Neural Networks** Radial basis function networks were introduced by Broomhead & Lowe (1988). The networks formulate activation functions as RBFs, resulting in an output that is a sum of radial basis function values between the input and network parameters. In contrast to these radial basis function networks, our approach uses RBFs in the solver of a deep convolutional neural network and our radial basis function centres are coupled to high dimensional embeddings of training samples, rather than being network parameters. Radial basis functions have been used as neural network solvers in the form of support vector machines. In one such formulation, a neural network is used as a fixed feature extractor and separate support vector machines are trained to classify the features (Razavian et al., 2014; Donahue et al., 2014). No joint training occurs between the solver (classifier) and network. Such an approach is often used for transfer learning, where the network is trained on vast amounts of data and the support vector machines are trained for problems in which labelled training data is scarce. Tang (2013) replaces the typical softmax classifier with linear support vector machines. In this case, the solver and network are trained jointly, meaning the loss that is minimised is margin based.

**Metric Learning** Early methods in the domain of metric learning include those that use Siamese networks (Bromley et al., 1993) and contrastive loss (Hadsell et al., 2006; Chopra et al., 2005). The objective of these approaches is to pull pairwise samples of the same class together and push pairwise samples of different classes apart. Such methods work on absolute distances, while triplet networks with hinge loss (Weinberger et al., 2006) work on relative distance. Triplet loss approaches take a trio of inputs; an anchor, a positive sample of the same class as the anchor and a negative sample of a different class. Triplet loss aims to pull the positive sample closer to the anchor than the negative sample. Several deep metric learning approaches make use of, or generalise deep triplet

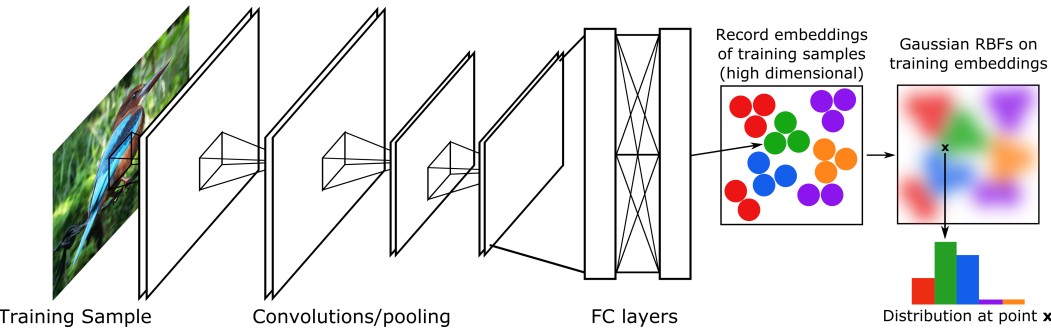

Figure 1: Overview of our radial basis function solver.

neural networks (Hoffer & Ailon, 2015; Wang et al., 2014; Schroff et al., 2015; Song et al., 2016a; Sohn, 2016; Kumar et al., 2017). Schroff et al. (2015) perform semi-hard mining within a mini-batch, while Song et al. (2016a) propose a lifted structured embedding with efficient computation of the full distance matrix within a mini-batch. This allows comparisons between all positive and negative pairs in the batch. Similarly, Sohn (2016) proposes an approach that allows multiple intra-batch distance comparisons, but optimises a generalisation of triplet loss, named N-pair loss, rather than a max-margin based objective, as in Song et al. (2016a). The global embedding structure is considered in Song et al. (2016b) by directly minimising a global clustering metric, while a combination of global and triplet loss is shown to be beneficial in Kumar et al. (2016). Finally, Kumar et al. (2017) introduce a smart mining technique that mines for triplets over the entire dataset. A Fast Approximate Nearest Neighbour Graph (FANNG) (Harwood & Drummond, 2016) is leveraged for computational efficiently. Beyond triplet loss, Rippel et al. (2016) introduce a loss function that allows multiple clusters to form per class. Rather than only penalising a single triplet at a time, the neighbourhood densities are considered and overlaps between classes penalised.

## 3 RADIAL BASIS FUNCTION SOLVERS

A radial basis function returns a value that depends only on the distance between two points, one of which is commonly referred to as a centre. Although several radial basis functions exist, in this paper we use RBF to refer to a Gaussian radial basis function, which returns a value based on the Euclidean distance between a point $\mathbf{x}$ and the RBF centre $\mathbf{c}$. The radial basis function, $f$, is calculated as:

$$f(\mathbf{x}, \mathbf{c}) = \exp\left(\frac{-\|\mathbf{x} - \mathbf{c}\|^2}{2\sigma^2}\right) \qquad (1)$$

where $\sigma$ is standard deviation that controls the width of the Gaussian curve, that is, the region around the RBF centre deemed to be of importance.

In the context of our neural network solver, we define the deep feature embeddings of each training set sample as radial basis function centres. Specifically, we take the layer in a network immediately before the solver as the embedding layer. For example, in a VGG architecture, this may be FC7 (fully connected layer 7), forming a 4096 dimension embedding. In general, however, the embedding may be of any size. An overview of this approach is seen in Figure 1.

### 3.1 CLASSIFIER AND LOSS FUNCTION

A radial basis function classifier can be formed by the weighted sum of the RBF distance calculations between a sample feature embedding and the centres. Classification of a sample is achieved by passing the input through the network, resulting in a feature embedding in the same space as the RBF centres. A probability distribution over class labels is found by summing the influence of each centre and normalising. A centre contributes only to the probability of the ground truth label of the training sample coupled to that centre. For example, the probability that the feature embedding $\mathbf{x}$ has class label $Q$ is:

$$Pr(\mathbf{x} \in \text{class } Q) = \frac{\sum_{i \in Q} w_i f(\mathbf{x}, \mathbf{c_i})}{\sum_{j=1}^{m} w_j f(\mathbf{x}, \mathbf{c_j})}, \qquad (2)$$

where $f$ is the RBF, $i \in Q$ are the centres with label $Q$, $m$ is the number of training samples and $w_i$ is a learnable weight for RBF centre $i$. Of course, if a sample is in the training set and has a corresponding RBF centre, the distance calculation to itself is omitted during the computation of the classification distribution, the loss function and the derivatives.

The loss function used for optimisation is simply the summed negative logarithm of the probabilities of the true class labels. For example, the loss $L$ for sample $\mathbf{x}$ with ground truth label $R$ is:

$$L(\mathbf{x}) = -\ln\left(Pr(\mathbf{x} \in \text{class } R)\right). \tag{3}$$

The same loss function is used regardless of whether the network is being trained for classification, as above, or for embedding space building (distance metric learning). This is possible since the RBF classifier is directly computed from distances between features in the embedding space. This means that a network trained for classification will result in features of the same class being located near one another, and similarly a network trained for metric learning will result in an embedding space in which features can be well classified using RBFs.

## 3.2 Nearest Neighbour RBF Solver

In Equation 2, the distribution is calculated by summing over all RBF centres. However, since these centres are attached to training samples, of which there could be any large number, computing that sum is both intractable and unnecessary. The majority of RBF values for a given feature embedding will be effectively zero, as the sample feature will lie only within a subset of the RBF centres' Gaussian windows. As such, only the local neighbourhood around a feature embedding should be considered. Operating on the set of the nearest RBF centres to a feature ensures that most of the distance values computed are pertinent to the loss calculation. The classifier equation becomes:

$$Pr(\mathbf{x} \in \text{class } Q) = \frac{\sum_{i \in Q \cap \mathcal{N}} w_i f(\mathbf{x}, \mathbf{c_i})}{\sum_{j \in \mathcal{N}} w_j f(\mathbf{x}, \mathbf{c_j})}, \tag{4}$$

where $\mathcal{N}$ is the set of approximate nearest neighbours for sample $\mathbf{x}$ and therefore $i \in Q \cap \mathcal{N}$ is the set of approximate nearest neighbours that have label $Q$. Again, we note that training set samples exclude their own RBF centre from their nearest neighbour list.

In the interest of providing a scalable solution, we use approximate nearest neighbour search to obtain candidate nearest neighbour lists. This allows for a trade off between precision and computational efficiency. Specifically, we use a Fast Approximate Nearest Neighbour Graph (FANNG) (Harwood & Drummond, 2016), as it provides the most efficiency when needing a high probability of finding the true nearest neighbours of a query point. Importantly, FANNG provides scalability in terms of the number of dimensions and the number of training samples.

## 3.3 End-to-end Learning

The network and solver weights are learned end-to-end. As the weights are constantly being updated during training, the locations of the RBF centres are changing. This leads to complications in the computation of the derivatives of the loss with respect to the embeddings. This calculation requires dimension by dimension differences between the training embeddings and the RBF centres. The centres are moving as the network is being updated, but computing the current RBF centre locations online is intractable. For example, if considering 100 nearest neighbours, 101 samples would need to be forward propagated through the network for each training sample. However, we find that is is not necessary for the RBF centres to be up to date at all times in order for the model to converge. A bank of the RBF centres is stored and updated at a fixed interval.

A further consequence of the RBF centres moving during training is that the nearest neighbours also change. It is intractable to find to correct nearest neighbours each time the weights are updated. This is simply remedied by considering a larger number of nearest neighbours than would be required if all centres and neighbour lists were up-to-date at all times. The embedding space changes slowly enough that it is highly likely many of the previously neighbouring RBF centres will remain relevant. Since the Gaussian RBF decays to zero as the distance between the points becomes large, it does not matter if an RBF centre that is no longer near the sample remains a candidate nearest neighbour.

We call the frequency at which the RBF centres are updated and the nearest neighbours found the *update interval*. During training, at a fixed number of epochs we forward pass the entire training set

through the network, storing the new RBF centres. The up-to-date nearest neighbours can now be found. If FANNG is used, a rebuild of the graph is required. Note that the stored RBF centres do not have dropout (Srivastava et al., 2014) applied, but the current training embeddings may. The effect of the number of nearest neighbours considered and the update interval are discussed in Section 4.2.

**Radial Basis Function Parameters**   A global standard deviation parameter $\sigma$ is shared amongst the RBFs. This ensures that the assumption made about samples only being influenced by their nearest RBF centres holds. Although the parameter is learnable, we find that fixing the standard deviation value before training is a suitable approach. We treat the standard deviation as an additional hyperparameter to tune, however it can also be learned independently before full network training commences. As seen in Equation 4, each RBF centre has a weight, which is learned end-to-end with the network weights. These weights are initialised at values of one. Note that in our experiments we only tune the RBF weights for the classification task; they remain fixed for metric learning problems.

## 4   EXPERIMENTS

We detail our experimental results in two tasks; distance metric learning and image classification.

### 4.1   DISTANCE METRIC LEARNING

**Experimental Set-up**   We evaluate our approach on two datasets; Stanford Cars196 (Krause et al., 2013) and CUB-200-2011 (Birds200) (Welinder et al., 2010). Cars196 consists of 16,185 images of 196 different car makes and models, while Birds200 consists of 11,788 images of 200 different bird species. In this problem, the network is trained and evaluated on different sets of classes. We follow the experimental set-up used in Song et al. (2016a); Sohn (2016); Song et al. (2016b); Kumar et al. (2017). For the Cars196 dataset, we train the network on the first 98 classes and evaluate on the remaining 98. For the Birds200 dataset we train on the first 100 classes and evaluate on the remaining 100. Stochastic gradient descent optimisation is used. All images are first resized to be 256x256 and data is augmented by random cropping and horizontal mirroring. Note that we do not crop the images using the provided bounding boxes.

Our method is compared to state-of-the-art approaches on the considered datasets; semi-hard mining for triplet networks (Schroff et al., 2015), lifted structured feature embedding (Song et al., 2016a), N-pair loss (Sohn, 2016), clustering (Song et al., 2016b), global loss with triplet networks (Kumar et al., 2016) and smart mining for triplet networks (Kumar et al., 2017). For fair comparison to these methods, we use the same base architecture for our experiments; GoogLeNet (Szegedy et al., 2015). Network weights are initialised from ImageNet (Russakovsky et al., 2015) pre-trained weights. We use 100 nearest neighbours and an update interval of 10 epochs. RBF weights are fixed at a value of one for this task. We train for 50 epochs on Cars196 and 30 epochs on Birds200. A batch size of 20, base learning of 0.00001 and weight decay of 0.0002 are used. The RBF standard deviation used depends on size of the embedding dimension. We find values between 10 and 30 work well for this task.

**Evaluation Metrics**   Following Song et al. (2016a), we evaluate the embedding space using two metrics; Normalised Mutual Information (NMI) (Manning et al., 2008) and Recall@K. The NMI score is the ratio of mutual information and average entropy of a set of clusters and labels. It evaluates only for the number of clusters equal to the number of classes. As discussed in Section 1, a good embedding space does not necessarily have only one cluster per class, but may have multiple well formed clusters in the space. This means that our mutual information may be higher than reported with this metric. Nevertheless, we present results on the NMI score in the interest of comparing to existing methods that evaluate on this metric. The Recall@K (R@K) metric is better suited for evaluating an embedding space. A true positive is defined as a sample that has at least one of its true nearest K neighbours in the embedding space with the same class as itself.

**Embedding Space Dimension**   We investigate the importance of the embedding dimension. A similar study in Song et al. (2016a) suggests that the number of dimensions is not important for triplet networks, in fact, increasing the number of dimensions can be detrimental to performance. We compare our method with increasing dimension size against triplet loss (Weinberger et al., 2006;

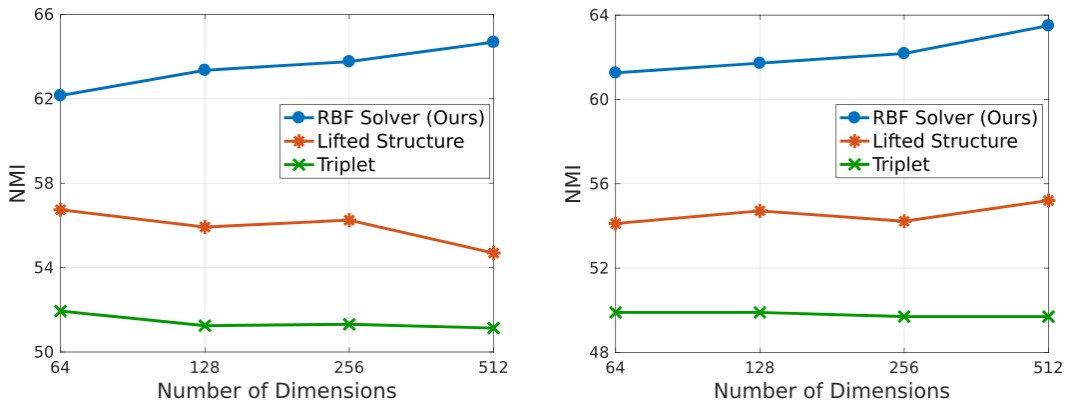

Figure 2: Effect of embedding size on NMI score on the test set of Cars196 (left) and Birds200 (right). The NMI of our RBF approach improves with increasing embedding size, while performance degrades or oscillates for triplet (Weinberger et al., 2006; Schroff et al., 2015) and lifted structured embedding (Song et al., 2016a).

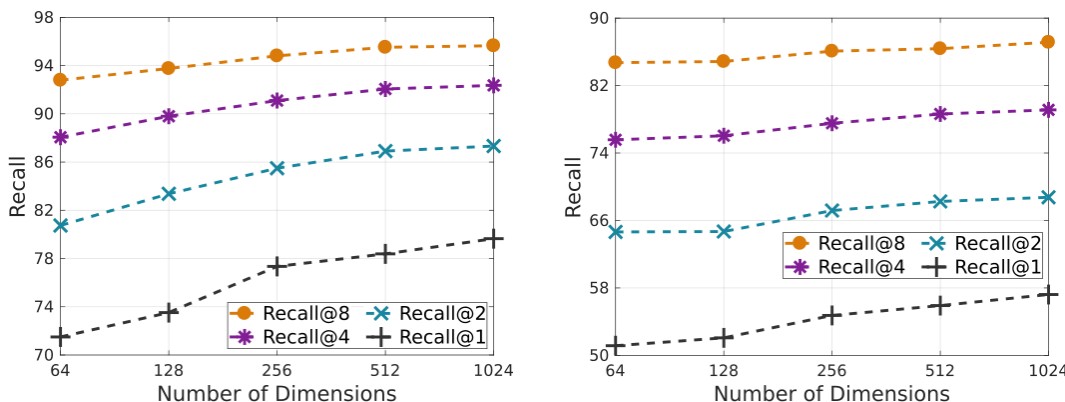

Figure 3: Recall of our RBF solver at 1, 2, 4 and 8 nearest neighbours on the test set of Cars196 (left) and Birds200 (right). Recall performance of our approach increases with embedding size.

Schroff et al., 2015) and lifted structured embedding (Song et al., 2016a), both taken from the study in Song et al. (2016a). Figure 2 shows the effect of the embedding size on NMI score. It's clear that while increasing the number of dimensions does not necessarily improve performance for triplet-based networks, the dimensionality is important for our RBF approach. The NMI score for our approach improves with increasing numbers of dimensions. Similar behaviour is seen in Figure 3, which shows the Recall@K metric for our RBF method with varying numbers of dimensions. Again, this shows that the dimensionality is an important factor for our approach.

**Comparison of Results**   Our approach is compared to the state-of-the-art in Table 1, with the compared results taken from Song et al. (2016b) and Kumar et al. (2017). Since, as discussed above, the number of embedding dimensions does not have much impact on the other approaches, all results in Song et al. (2016b) and Kumar et al. (2017) are reported using 64 dimensions. For fair comparison, we report our results at 64 dimensions, but also at the better performing higher dimensions. Our approach outperforms the other methods in both the NMI and Recall@K measures, at all embedding sizes presented. Our approach is able to produce better compact embeddings than existing methods, but can also take advantage of a larger embedding space. Figure 4 shows a t-SNE (van der Maaten & Hinton, 2008) visualisation of the Birds200 test set embedding space. Despite the test classes being withheld during training, bird species are well clustered.

Table 1: Embedding results on Cars196 and Birds200. The test set is comprised of classes on which the network was not trained. Our approach is compared with state-of-the-art approaches; Semi-hard (Schroff et al., 2015), LiftStruct (Song et al., 2016a), N-pairs (Sohn, 2016), Triplet/Gbl (Kumar et al., 2016), Clustering (Song et al., 2016b) and SmartMine (Kumar et al., 2017).

| | Dims | Cars196 Dataset | | | | | Birds200 Dataset | | | | |
| | | NMI | R@1 | R@2 | R@4 | R@8 | NMI | R@1 | R@2 | R@4 | R@8 |
|---|---|---|---|---|---|---|---|---|---|---|---|
| Semi-hard | 64 | 53.35 | 51.54 | 63.78 | 73.52 | 82.41 | 55.38 | 42.59 | 55.03 | 66.44 | 77.23 |
| LiftStruct | 64 | 56.88 | 52.98 | 65.70 | 76.01 | 84.27 | 56.50 | 43.57 | 56.55 | 68.59 | 79.63 |
| N-pairs | 64 | 57.79 | 53.90 | 66.76 | 77.75 | 86.35 | 57.24 | 45.37 | 58.41 | 69.51 | 79.49 |
| Triplet/Gbl | 64 | 58.20 | 61.41 | 72.51 | 81.75 | 88.39 | 58.61 | 49.04 | 60.97 | 72.33 | 81.85 |
| Clustering | 64 | 59.04 | 58.11 | 70.64 | 80.27 | 87.81 | 59.23 | 48.18 | 61.44 | 71.83 | 81.92 |
| SmartMine | 64 | 59.50 | 64.65 | 76.20 | 84.23 | 90.19 | 59.90 | 49.78 | 62.34 | 74.05 | 83.31 |
| RBF (Ours) | 64 | 62.15 | 71.05 | 80.74 | 88.06 | 92.79 | 61.26 | 51.15 | 64.64 | 75.57 | 84.72 |
| RBF (Ours) | 128 | 63.35 | 73.52 | 83.37 | 89.80 | 93.76 | 61.72 | 52.08 | 64.69 | 76.05 | 84.86 |
| RBF (Ours) | 256 | 63.76 | 77.35 | 85.49 | 91.10 | 94.81 | 62.18 | 54.74 | 67.18 | 77.53 | 86.09 |
| RBF (Ours) | 512 | 64.68 | 78.39 | 86.91 | 92.06 | 95.52 | 63.50 | 55.91 | 68.26 | 78.63 | 86.38 |
| RBF (Ours) | 1024 | **65.30** | **79.65** | **87.33** | **92.36** | **95.65** | **63.95** | **57.22** | **68.75** | **79.12** | **87.14** |

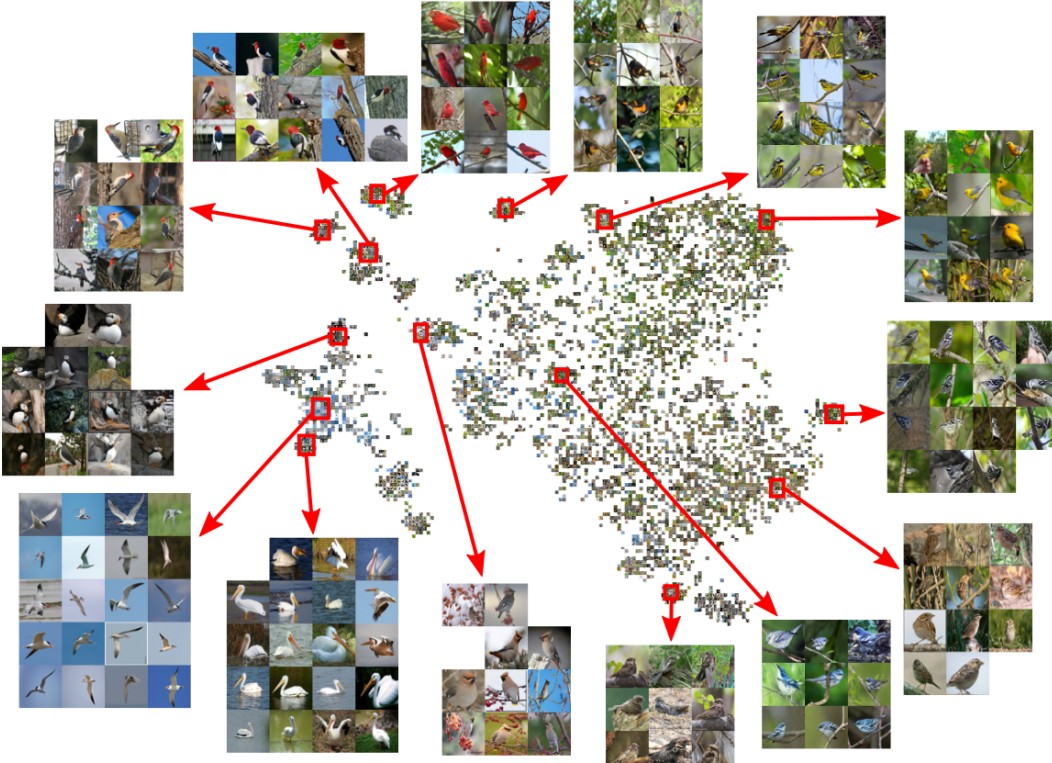

Figure 4: Visualisation of the Birds200 test set embedding space, using the t-SNE algorithm (van der Maaten & Hinton, 2008). Despite not being trained on the test classes, bird species are well clustered. Best viewed in colour and zoomed in on a monitor.

## 4.2 IMAGE CLASSIFICATION

**Experimental Set-up** We evaluate our solver in the domain of image classification, comparing performance with conventional softmax loss. For all experiments, images are resized to 256x256 and random cropping and horizontal mirroring is used for data augmentation. Unlike in Section 4.1, we crop Birds200 and Cars196 images using the provided bounding boxes before resizing. The same

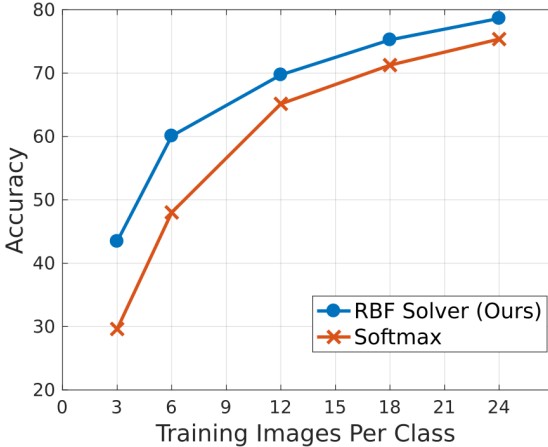

Figure 5: Effect of the number of training samples per class on the test set accuracy of Birds200, using a VGG16 architecture. Note that the final data point in the plot refers to the entire training set; while most classes have 24 training samples per class, some have only 23.

Table 2: Birds200 test set accuracy.

| | Solver | |
| --- | --- | --- |
| Base Network | Softmax | RBF (Ours) |
| AlexNet | 62.41 | **66.95** |
| VGG16 | 75.37 | **78.63** |
| ResNet50 | 78.05 | **78.98** |

classes are used for training and testing. All datasets are split in to training, validation and test sets. We select softmax and RBF hyperparameters that minimise the validation loss. The FC7 layer (4096 dimensions), with dropout and without a ReLU, is used as the embedding layer for our RBF solver when using a VGG (Simonyan & Zisserman, 2014) or AlexNet (Krizhevsky et al., 2012) architecture. For a ResNet architecture (He et al., 2016), we use the final pooling layer (2048 dimensions). We find that following the ResNet embedding layer with a dropout layer results in a small performance gain for both RBF and softmax solvers. A batch size of 20 is used and an update interval of 10 epochs, unless otherwise noted. We use stochastic gradient descent optimisation. In general, we find a base learning rate of 0.00001 to be appropriate for our approach. A standard deviation of around 100 for the RBFs is found to be suitable for the 4096 dimension VGG16 embeddings on Birds200. Networks are initialised with ImageNet (Russakovsky et al., 2015) pre-trained weights.

**Evaluation on Birds200** We carry out detailed evaluation of our approach on the Birds200 dataset. Since there is no standard validation set for this dataset, we take 20% of the training data as validation data. In Table 2, we evaluate with three network architectures; AlexNet (Krizhevsky et al., 2012), VGG16 (Simonyan & Zisserman, 2014) and ResNet50 (He et al., 2016). Our approach outperforms the softmax counterpart for each network. The performance gain over softmax is larger for AlexNet and VGG than for ResNet. This is likely because ResNet has significantly more non-linear activation function layers, meaning there is less improvement seen when using the highly non-linear RBF solver. The effect of the number of training samples per class is shown in Figure 5. Our RBF approach outperforms softmax loss at all numbers of training images, with a particularly large gain when training data is scarce.

Results from ablation experiments on our RBF approach are shown in Table 3. The importance of the following components of learning are shown; tuning the RBF standard deviation $\sigma$, learning the RBF weights and fine-tuning the network weights. Figure 6a shows the impact of the number of nearest neighbours used for each sample during training. There is a clear lower bound required for good performance. As discussed in Section 3.3, this is because the network weights are constantly

Table 3: Ablation study on Birds200.

| Initial Network Weights | Tune $\sigma$ | Learn RBF Weights | Fine-tune Network Weights | Test Accuracy |
|---|---|---|---|---|
| Random | Yes | No | No | 1.35 |
| ImageNet | Yes | No | No | 47.32 |
| ImageNet | Yes | Yes | No | 49.22 |
| ImageNet | Yes | No | Yes | 77.94 |
| ImageNet | Yes | Yes | Yes | 78.63 |

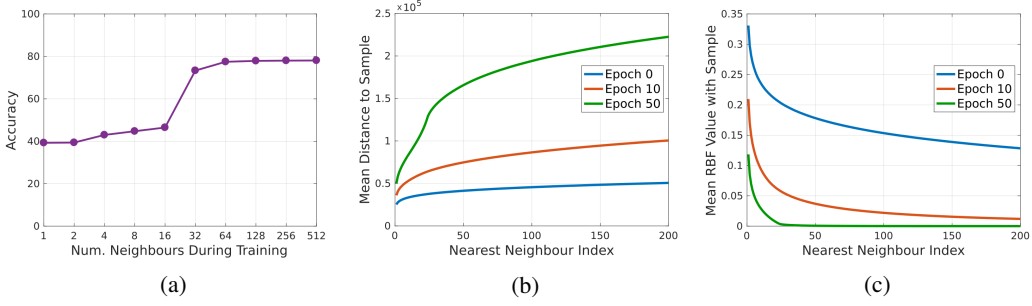

(a)          (b)          (c)

Figure 6: (a) The effect of the number of nearest neighbours considered during training. (b) The average distance from training samples to their nearest RBF centres. (c) The average RBF value between training samples and their nearest RBF centres.

being updated, but the stored RBF centres are not. As such, we need to consider a larger number of neighbours than if the centres were always up-to-date. Figure 6b shows the average distance from each training sample to its nearest RBF centres at different points during training. Similarly, Figure 6c shows the average radial basis function values between training samples and their nearest centres. These experiments use a VGG16 architecture.

When training with softmax loss on a VGG16 architecture, validation loss plateaus at around 7000 iterations. For our RBF solver, the number of iterations taken for validation loss to stop improving depends on the update interval, that is, the interval at which the RBF centres are updated and the nearest neighbours computed. For update intervals of 1, 5 and 10, validation loss stops improving at around 8500, 12000 and 15000 iterations, respectively. Since nearest neighbour search becomes the bottleneck as the dataset size increases, a less frequent update interval should be used for large datasets, allowing for a faster overall training time. The softmax solver is able to converge in fewer iterations than our approach. This is likely due to the RBF centres not being up-to-date at all times, leading to weight updates that are less effective than in the ideal scenario. However, as discussed in Section 3.3, keeping the RBF centres up-to-date at all times in intractable.

Our RBF approach allows clusters to position themselves freely in the embedding space, such that the intrinsic structure of the data can be represented. As a result, we expect the embeddings to be co-located based not only in terms of class, but also in terms of more fine-grained information, such as attributes. We use the 312 binary attributes of Birds200 to confirm this expectation. For each 4096 dimension VGG16 test set embedding, we propagate attributes by computing the density of each attribute label present in the neighbouring test embeddings. This is done using Gaussian radial basis functions, treating each attribute as a binary classification problem. We find the best Gaussian standard deviation for softmax and our RBF learned embeddings separately. A precision and recall curve, shown in Figure 7, is generated by sweeping the classification discrimination threshold from zero to one. We find that for a given precision, the RBF solver results in an embedding space with better attribute recall than softmax. Note that we do not train the models using the attribute labels.

**Other Datasets** We further evaluate our approach on three other fine-grained classification datasets; Oxford 102 Flowers (Nilsback & Zisserman, 2008), Stanford Cars196 (Krause et al., 2013) and

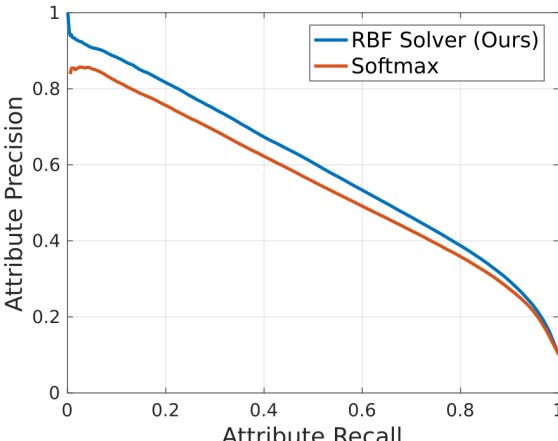

Figure 7: Attribute precision and recall on the 312 binary attributes of Birds200. The attributes are propagated from neighbouring test embeddings and the curves are generated by sweeping the classification discrimination threshold. The ideal standard deviation is found for the RBF and softmax approaches separately. No training was carried out on the attribute labels.

Table 4: Test accuracy on fine-grained classification datasets.

| Dataset | Softmax | RBF (Ours) |
|---|---|---|
| Oxford 102 Flowers | 82.79 | **86.26** |
| Stanford Cars196 | 85.67 | **86.52** |
| Leafsnap Field | 73.80 | **75.96** |

Leafsnap (Kumar et al., 2012). We use the standard training, validation and test splits for Oxford 102 Flowers. For Stanford Cars196, we take 30% of the training set as validation data. We use the challenging *field* images from Leafsnap, which are taken in uncontrolled conditions. The dataset contains 185 classes of leaf species and we split the data into 50%, 20% and 30% for training, validation and testing, respectively. Again, hyperparameters are selected based on validation loss and a VGG16 architecture is used. Results are shown in Table 4.

## 5  DISCUSSION AND CONCLUSION

Our approach is designed to address two problems; metric space learning and classification. The use of RBFs arises very naturally in the context of the first problem because metric spaces are defined and measured in terms of Euclidean distance. It is perhaps more surprising that the classification problem also benefits from using a metric space kernel density approach, rather than softmax. This appears to hold independently of the base network architecture (Table 2) and the improvement is particularly strong when limited quantities of training data are available (Figure 5).

Metric learning inherently pulls samples together into high density regions of the embedding space, whereas softmax is content to allow samples to fill a very large region of space, provided that the logit dimension corresponding to the correct class is larger than the others. This suggests that metric learning is able to provide some regularisation, because classification is driven by multiple nearby samples, whereas samples may be well separated in logit space for softmax. In turn, this leads to increased robustness for the metric space approach, particularly when training data is impoverished. Additionally, softmax is constrained to push samples into regions of space determined by the locations of the logit axes, whereas our metric learning approach is free to position clusters in a way that may more naturally reflect the intrinsic structure of the data. Finally, our approach is also free to create multiple clusters for each class, if this is appropriate. As a result of these factors, our RBF solver is able to outperform state-of-the-art approaches in the metric learning problem, as well as provide benefit over softmax in the classification problem.

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
