# OpenReview forum: "Nearest Neighbour Radial Basis Function Solvers for Deep Neural Networks"
_ICLR.cc/2018/Conference — Reject_

### Official Review · AnonReviewer3 · 2017-11-27

**Rating:** 5
**Confidence:** 4

**Review:**

(Summary)
This paper proposes weighted RBF distance based loss function where embeddings for cluster centroids and data are learned and used for class probabilities (eqn 3). The authors experiment on CUB200-2011, Cars106, Oxford 102 Flowers datasets.

(Pros)
The citations and related works cover fairly comprehensive and up-to-date literatures on deep metric learning.

(Cons)
The proposed method is unlikely to scale with respect to the number of classes. "..our approach is also free to create multiple clusters for each class.." This makes it unfair to deep metric learning baselines in figures 2 and 3 because DMP baselines has memory footprint constant in the number of classes. In contrast, the proposed method have linear memory footprint in the number of classes. Furthermore, the authors ommit how many centroids are used in each experiments.

(Assessment)
Marginally below acceptance threshold. The method is unlikely to scale and the important details on how many centroids the authors used in each experiments is omitted.

---

> ### Author Response · Authors · 2017-12-23
> **Response to Reviewer 3**
>
> Thank you for your comments. The main points of your review are addressed below.
>
> -Scalability with the number of classes.
>
> Scalability in terms of computation: As the number of classes (and therefore the number of training examples) increases, the number of RBF centres also increases. Our approach is scalable to large numbers of training examples, as we use fast approximate nearest neighbour search to obtain approximate nearest neighbour subsets for computing the loss and gradients. Fast Approximate Nearest Neighbour Graphs (Harwood and Drummond, 2016) make our approach scalable up to a very large number of training set examples, as well as a large embedding dimension.
>
> Scalability in terms of performance: Figure 4 suggests that the margin between our approach and softmax classification performance will shrink as the number of training examples per class becomes larger. However, there is nothing to suggest that our approach will scale poorly as the number of classes increases.
>
>
> -Memory footprint of DML approaches and fair comparison.
>
> The memory footprint of our approach during training is linear with the number of training set examples, not the number of classes. This is the same for the DML baselines, which record each training set embedding in order to collect statistics and perform triplet selection. For example, Kumar et al. (2017) perform smart mining over all of the training set embeddings to select triplets. As such, our comparisons to the DML baselines are fair.
>
>
> -Number of centroids used.
>
> The number of centroids is not a hyperparameter of our model. Our approach is free to form as many clusters for a given class as best represents the data. This is an advantage of our approach compared to other deep metric learning approaches that attempt to learn local similarity, since we do not have to determine the desired number of clusters or the cluster size before training.
>
> Thank you again for your review.

---

### Official Review · AnonReviewer2 · 2017-11-27
**The paper proposes a non-linear RBF kernel based layer on top of standard CNNs for improving classification. The presentation quality is not good, and the novelty is low.**

**Rating:** 3
**Confidence:** 4

**Review:**

- The paper proposes to use RBF kernel based neurons with each training data point as a center of
  one of the RBF kernel neuron. (i) Kernel based neural networks have been explored before [A] and
  (ii) ideas similar to the nearest neighbour based efficient but approximate learning for mixture
  of Gaussians like settings have also been around, e.g. in traning GMMs [B]. Hence I would consider
  the novelty to be very low
- The paper says that the method can be applied to embedding learning and classification, which were
  previously separate problems. This is largely incorrect as many methods for classification,
  especially in zero- and few-shots settings (on some of the datasets used in the paper) are using
  embedding learning [C], one of the cited and compared with paper (Sohn 2016) also does both
  (mostly these methods use k-NN classifier with Euclidean distance between learned embeddings)
- It seems that the method thus is adding a kernel neuron layer, with the number equal to the number
  of training samples, centers initialized with the training samples, followed by a normalized
  voting based on the distance of the test example with training examples of different classes
  (approximately a weighted k-NN classifier)
- The number of neurons in the last layer thus scales with the number of training examples, which
  can be prohibitively large
- It is difficult to understand what exactly is the embedding; if the number of neurons in the
  RBF layer is equal to the number of training examples then it seems the embedding is the activation
  of the layer before that (Fig1 also seems to suggest this). But the evaluation is done with
  different embedding sizes, which suggests that another layer was inserted between the last FC
  layer of the base network and the RBF layer. In that case the empirical validation is not fair as
  the network was made deeper.
- Also, it is a bit confusing that as training proceeds the centers change (Sec3.3 first few lines),
  so the individual RBF neurons, eventually, do not necessarily correspond to the training examples
  they were initialized with, but the final decision (Eq4) seems to be taken assuming that the
  neurons do correspond to the training examples (and their classes). While the training might
  ensure that the centers do not move so much, this should be explicitly discussed and clarified.

Overall, the novelty of the paper seems to be low and it is difficult to understand what exactly is
being done.

[A] Xu et al., Kernel neuron and its training algorithm, ICONIP 2001
[B] Verbeek et al., Efficient greedy learning of gaussian mixture models, Neural Computation 2003
[C] Xian et al., Latent Embeddings for Zero-shot Classification, CVPR 2016

---

> ### Author Response · Authors · 2017-12-23
> **Response to Reviewer 2**
>
> Thank you for your comments and review. The major points raised are addressed below.
>
> -Kernel based neurons.
>
> Kernel based neurons have been explored before and we discuss this briefly in our literature review (such as the work by Broomhead and Lowe (1988)). Unlike kernel neuron approaches, our RBF neurons are not learnable parameters of the model. Rather, the RBF centres are coupled to high dimensional training set feature embeddings. Further, we introduce a learnable per exemplar weight for the RBF centres. We also show how to make training tractable by allowing RBF centres to become out-of-date with the training embeddings for periods of time during training. Finally, we demonstrate how to make our approach scalable with the number of training examples and the embedding dimension, by leveraging fast approximate nearest neighbour search.
>
>
> -Other approaches for both classification and metric learning.
>
> As stated, embedding learning approaches have been used for zero or few shot classification scenarios, but these approaches do not scale well beyond these settings of impoverished training data. This is seen in the comparison between triplet and softmax for classification in Rippel et al. (2016). On the same and similar datasets as experimented with in our paper, triplet deep metric learning approaches under perform softmax for classification tasks by up to 10% (Rippel et al. 2016). Contrary to this, our metric learning approach outperforms softmax on such datasets. Although there are a few other examples of metric learning approaches that have been used for classification, such as in Sohn (2016), the advantage over a softmax classifier is inconsistent between datasets.
>
>
> -The embedding and fair comparison to other metric learning approaches.
>
> No extra depth is added to the base networks to which we compare. In the classification experiments, the softmax baseline networks have a final FC layer, with the number of channels equal to the number of classes, and softmax loss applied to the output of this layer. Our approach removes the softmax and final FC layer and replaces them with our RBF loss layer. This means the embedding is the output from the layer immediately before the final FC layer in the softmax network (e.g. the 4096 dimension FC7 for VGG16 or the 2048 dimension final average pooling layer for ResNet). As such, the model capacity of our approach is reduced compared to softmax.
>
> For the comparison to deep metric learning approaches (Table 1), we follow the exact same set-up used in the papers to which we compare. These approaches insert an additional FC layer after the final average pooling layer of GoogLeNet, in order to achieve the desired embedding dimension. The compared approaches use an embedding dimension of 64 and we show that our approach outperforms these methods at this sized embedding. Additionally, we show that our approach is able to take advantage of a larger embedding space, while triplet based approaches do not see the same benefit from increasing the embedding dimension (Figures 2 and 3).
>
>
> -RBF centres moving during training.
>
> The centres are updated at regular intervals during training (every 1, 5 or 10 epochs, for example). This is shown to be sufficient for the model to learn. Any testing of the network is carried out with fully updated centres (i.e. centres that correspond exactly to the training examples). Practically, this is done by doing a full forward pass of the training data at the predefined interval during training and updating the model parameters that correspond to the RBF centres. At the completion of training, the centres are again updated to correspond with the training examples, before testing/deploying the model.
>
> Thank you again for your comments.

---

### Official Review · AnonReviewer1 · 2017-11-28
**Rejection of paper with potential if reframed as revisiting the NCA loss.**

**Rating:** 4
**Confidence:** 4

**Review:**

The authors propose a loss that is based on a RBF loss for metric learning and incorporates additional per exemplar weights in the index for classification. Significant improvements over softmax are shown on several datasets.

IMHO, this could be a worthwhile paper, but the framing of the paper into existing literature is lacking and thus it appears as if the authors are re-inventing the wheel (NCA loss) under a different name (RBF solver).

The specific problems are:
- The authors completely miss the connection to NCA loss (https://papers.nips.cc/paper/2566-neighbourhood-components-analysis.pdf) and thus appear to be re-inventing the wheel.
  - The proposed metric learning scenario is exactly as proposed in the NCA loss works, while the classification approach adds an interesting twist by learning per exemplar weights. I haven't encountered this before and it could make an interesting proposal. Of course the benefit of this should be evaluated in ablation studies( Tab 3 shows one experiment with marginal improvements).
- The authors' use of 'solver' seems uncommon and confusing. What is proposed is a loss in addition to building a weighted index in the case of classification.
- In the metric learning comparison with softmax (end of page 9) the authors mentions that a Gaussian standard deviation for softmax is learned. It appears as if the authors use the softmax logits as embedding whereas the more common approach is to use the bottleneck layer. This is also indicated by the discussion at the end of page 10 where the authors mention that softmax is restricted to axis aligned embeddings. All softmax metric learning experiments should be carried out on appropriately sized bottleneck layers.
- Some of the motivations of what the various methods learn seem flawed, e.g. triplet loss CAN learn multiple modes per class and there is nothing in the Softmax loss that encourages the classes to fill a large region of the space.
- Why don't the authors compare on ImageNet?

Some positive points:
- The authors mention in Sec 3.3 that updating the RBF centres is not required. This is a crucial point that should be made a centerpiece of this work, as there are many metric learning works that struggle with this. Additional experiments that can investigate this point would greatly contribute to a well rounded paper.
- The numbers reported in Tab 1 show very significant improvements

If the paper was re-framed and builds on top of the already existing NCA loss, there could be valuable contributions in this paper. The experimental comparisons are lacking in some respect, as the comparison with Softmax as a metric learning method seems uncommon, i.e. using the logits instead of the bottleneck layer. I encourage the authors to extend the paper and flesh out some of the experiments and then submit it again.

---

> ### Author Response · Authors · 2017-12-23
> **Response to Reviewer 1**
>
> Thank you for review and suggestions to improve the work. We address the main points of your review below.
>
> -NCA loss.
>
> There is indeed a strong connection between our work and NCA loss, and we thank the reviewer for pointing us towards this missed reference. However, this does not detract from the following novel contributions of our paper. Firstly, our approach is scalable both in the number of training examples and the embedding dimension, due to the leveraging of fast approximate nearest neighbour search. Further, our approach is contextualised amongst current deep metric learning approaches and applied to the domains of transfer learning and classification. We show how to train a deep neural network using our loss function to achieve state-of-the-art transfer learning/embedding space learning results, while also outperforming softmax-based classification. These two different target domains are rarely tackled simultaneously in the literature. Additionally, our approach addresses the issues associated with nearest neighbour based learning, by allowing RBF centres to become out-of-date with the training embeddings. We further perform an analysis on the number of nearest neighbours required for the model to learn, when the centres and training embeddings drift apart. Unlike the linear transformation in NCA, our approach learns a non-linear transformation from the input space to the embedding space. We also study the importance of the embedding dimension, which is not addressed in the NCA work. Finally, our approach includes a learnable weight per exemplar, strengthening the classification capability of the model.
>
>
> -Clarification on using bottleneck layer for softmax.
>
> We do in fact use the bottleneck layer for these experiments, not the softmax logits. The FC7 layer of VGG16, with a 4096 dimension output, is used for both our approach and the softmax metric learning experiments.
>
>
> -Triplet loss and multiple modes per class.
>
> As triplet loss demands that semantically related instances are located nearby, and the only form of supervisory semantic information used is the class labels, standard triplet loss approaches will attempt to form a single cluster per class. The local structure of the space isn’t considered, meaning that any notion of intra-class similarity is lost. Although there are some approaches that attempt to represent local similarity, these require the parameters of the embedding space, such as the number of modes per class or the cluster size, to be determined before training. This is not ideal as this information cannot be determined by simply looking at the input space. Our approach, however, makes no assumptions about how the embedding space should be structured and allows clusters to form freely.
>
> Thank you again for your comments.

---

### Decision · Program_Chairs · 2018-01-29
**ICLR 2018 Conference Acceptance Decision**

**Decision:**

Reject

**Comment:**

This paper proposes a non-parametric method for metric learning and classification. One of the reviewers points out that it can be viewed as an extension of NCA. There is in fact a non-linear version of NCA that was subsequently published, see [1]. In this sense, the approach here appears to be a version of nonlinear NCA with learnable per-example weights, approximate nearest neighbour search, and the allowance of stale exemplars. In this view, there is concern from the reviewers that there may not be sufficient novelty for acceptance.

The reviewers have concerns with scalability. It would be helpful to include clarification or even some empirical results on how this scales compared to softmax. It is particularly relevant for larger datasets like Imagenet, where it may be impossible to store all exemplars in memory.

It is also recommended to relate this approach to metric-learning approaches in few-shot learning. Particularly to address the claim that this is the first approach to combine metric learning and classification.

[1]: Learning a Nonlinear Embedding by Preserving Class Neighbourhood Structure. Ruslan Salakhutdinov and Geoffrey Hinton.  AISTATS 2007